# 2,5-Furandicarboxylic Acid: An Intriguing Precursor for Monomer and Polymer Synthesis

**DOI:** 10.3390/molecules27134071

**Published:** 2022-06-24

**Authors:** Adam Marshall, Bo Jiang, Régis M. Gauvin, Christophe M. Thomas

**Affiliations:** Institut de Recherche de Chimie Paris, CNRS, Chimie ParisTech, PSL University, 75005 Paris, France; adammarshall15@outlook.com (A.M.); jb20798673@gmail.com (B.J.)

**Keywords:** biobased monomers, sustainable polymers, furan-based materials

## Abstract

The most versatile furanic building block for chemical and polymer applications is 2,5-furandicarboxylic acid. However, the classical 2,5-furandicarboxylic acid production methodology has been found to have significant drawbacks that hinder industrial-scale production. This review highlights new alternative methods to synthesize 2,5-furandicarboxylic acid that are both more advantageous and attractive than conventional oxidation of 5-hydroxymethylfurfural. This review also focuses on the use of 2,5-furandicarboxylic acid as a polymer precursor and the various potential applications that arise from these furan-based materials.

## 1. Introduction

Negative environmental impacts and the depletion of fossil sources are strong motivations for reducing the global dependence on fossil resources for plastics production. Currently, about 99% of all plastics produced are petroleum-based [1]. Approximately 4–6% of European oil is used to produce plastics [2], 40% of which can be attributed to the packaging industry [3]. Hence, the development of sustainable processes for renewable bioplastics for packaging offers the potential to replace around 2% of oil consumption. However, at present, only about 1% of the 360 million tons of plastic produced each year are bioplastics. This equates to around 2.11 million tons of bioplastics in 2019, 53% of which were produced for packaging applications [1]. So, here we have growing market interest and room for innovation to develop more sustainable bioplastics production.

The U.S. Department of Energy designated 2,5-furandicarboxylic acid (FDCA) as one of the “Top Value-Added Chemicals from Biomass” [4]. The production of FDCA has been studied since the 19th century [5]; however, due to low product selectivity and poor yields, as well as uneconomical reaction conditions, many of these methods proved unsuitable for industrial applications [6]. From a circular economy perspective, more recent studies on the synthesis of FDCA have involved the valorization of lignocellulosic materials, which can be derived from agricultural and food waste streams [7]. High-value C6-bifunctionalized furan-derived compounds can be synthesized from the C5 or C6 sugars present in lignocellulosic biomass via two key platform chemicals, namely furfural and 5-hydroxymethylfurfural (HMF) [8,9]. This new interest in FDCA production has been driven by the growing demand for sustainable bio-derived plastics.

FDCA bears a structural resemblance to terephthalic acid, a petroleum-derived monomer used to make plastics for food and beverage packaging (Figure 1). Terephthalic acid is industrially polymerized with ethylene glycol to obtain the commodity polymer poly(ethylene terephthalate) (PET), which is produced on a scale of 50 million tons per year. As PET is widely used in packaging, an FDCA-based polymer alternative must meet the same high standards of stability, mechanical strength, color, transparency, and gas barrier properties, if it is to compete as a renewable alternative [10].

Although some companies are taking steps toward commercial FDCA production, conventional synthetic routes still present many drawbacks that hinder industrial applications, as will be detailed below. In addition, the main application of FDCA for the synthesis of poly(ethylene furanoate) (PEF) has not been directly compared to the alternative polyesters that can be produced from the parent diol of FDCA. In this review article, we will present recent developments in the production and use of FDCA. This includes a review of the current routes and methods of producing FDCA and its products, in order to understand the problems encountered with these approaches and to propose beneficial alternatives. Alternatives to FDCA have also been studied, namely the corresponding diol and dimethyl ester. Methods and products from these derivatives were extensively reviewed to determine whether FDCA is the best choice for the final product. Finally, advances toward industrial FDCA production were examined to understand the challenges faced by manufacturers, both technically and economically. The overall objective of this article is therefore to review current and emerging routes to and from FDCA—that is, new routes to FDCA synthesis—and its potential applications using the many available products of FDCA and its associated esters and diols.

## 2. FDCA Synthesis

The most common method for the synthesis of FDCA from lignocellulosic biomass is the catalytic oxidation of HMF [8]. Despite a large number of studies on the process, this route is still not economically feasible on an industrial scale due to the cost of the process along with technical issues such as the low efficiency in the production and isolation of HMF from lignocellulosic biomass and the poor selectivity to FDCA [11,12]. Lignocellulosic biomass comes from waste streams such as agriculture, forestry, and paper, and therefore does not compete with food sources. Lignocellulose itself is a composite of three types of materials: lignin (an aromatic polymer), cellulose, and hemicellulose (hexose and pentose polysaccharides) [13]. The acid-catalyzed hydrolysis of cellulose and hemicellulose produces glucose, fructose, and xylose, which can be dehydrated to produce the platform chemicals HMF and furfural. These compounds are furanic molecules functionalized with formyl- and hydroxymethyl groups that can be converted to a wide range of highly valuable C6-bifunctionalized furanic products.

The classical oxidation of HMF to FDCA occurs via an easy pathway (Figure 1). Once HMF has been produced from C6 sugars by acid hydrolysis, a series of oxidation steps are performed before the formation of FDCA. If the alcohol function on HMF is oxidized first, 2,5-diformylfuran (DFF) will be produced. If the aldehyde group is oxidized first, the intermediate will instead be 5-hydroxymethylfuran-2-carboxylic acid (HMFCA). Then, either compound is oxidized to 5-formylfuran-2-carboxylic acid (FFCA), and finally to FDCA [13]. There are two main problems in this regard. First, the hydrolysis of cellulose to HMF produces insoluble polymeric by-products called humins, which color the final product and are difficult to remove. Second, the reaction may also involve further oxidation products of FDCA such as CO_2_ and CO, which means that at the end of the reaction, the selectivity to FDCA is low, as other products have been formed by decarboxylation [14]. This is a challenging reaction that can either achieve complete HMF conversion, resulting in a mixture of FDCA with its oxidation products (e.g., CO_2_, CO), or stop the reaction before this additional oxidation, resulting in a mixture of all of the intermediates, with low conversion to FDCA. In this section, we will review new methods that can address this problem, either directly by solving the issues of HMF oxidation, or by presenting alternative routes to FDCA that avoid these issues altogether. As shown in Figure 2, these new pathways can be divided into two sections (to the left and right of the lignocellulosic biomass on the diagram): routes from cellulose (glucose and fructose) and routes from hemicellulose (xylose).

### 2.1. FDCA from Cellulose

Avantium, a Dutch renewable chemistry company, has developed and patented a method for the synthesis of aromatic dicarboxylic acids by electrolytic oxidation of an aqueous electrolyte feedstock containing aromatic aldehydes using non-noble metal electrodes [15]. This method facilitates the oxidation of aromatic aldehyde FFCA to FDCA. Avantium reports that previous studies on the electrolytic oxidation of HMF directly to FDCA showed poor yields and that the conversion of the aldehyde functional group (as in FFCA) to a carboxylic functional group is “easier” than the oxidation of a furfural derivative with a hydroxymethyl group (as in HMF) [15]. The results claim complete conversion of the starting materials to FDCA with this method. Feedstocks containing only the aromatic aldehyde require considerable residence times, making the industrial process uneconomical. The process can be improved by including an aromatic dicarboxylic acid in the feedstock, preferably the same as the product to be generated (i.e., FDCA). When the aromatics in question are furanic derivatives, the process will also oxidize other components of the feedstock, such as HMF, DFF, and HMFCA. The application of this process can then be used for the purification of the products of conventional oxidation. It will oxidize the intermediates of conventional oxidation without the formation of CO_2_ and CO, thus increasing the yield of FDCA. This negates the high yield requirement in the conventional oxidation process, as all intermediates can be converted to FDCA if the feedstock contains up to 10 wt% FFCA. Furthermore, during conventional oxidation, humin by-products act as colorants that must be removed from the FDCA crude by additional separation methods. Avantium’s electrolytic oxidation process also removes humins, saving money by eliminating the need for more separation processes. In addition, there is no requirement to use expensive electrodes. The cathode can be made from carbon and the anode from non-noble metals or their oxides/hydroxides on a carbon support.

Despite extensive research, the production of the HMF platform chemical from abundant lignocellulosic biomass has been found to be uneconomical on an industrial scale in most cases [12]. Kang et al. proposed 5-acetoxymethylfufural (AMF) as a suitable alternative to HMF, as it can also be derived from cellulose and offers simple pathways to furan derivatives, including FDCA [16]. As discussed earlier, HMF is produced by acidic dehydration of the carbohydrates in lignocellulosic biomass, a reaction that involves many side reactions. By-products, including levulinic acid and insoluble polymeric humins, are formed during the hydrolysis and condensation of HMF, which reduces the selectivity and efficiency of the process [6]. In contrast, AMF can be easily synthesized from lignocellulosic biomass-derived 5-chloromethylfurfural (CMF) and alkylammonium acetates in a process without side reactions and/or by-products (Figure 2). The acetoxymethyl group in AMF makes the compound less reactive and more hydrophobic than the hydroxymethyl group in HMF. This stability (and hydrophobicity) facilitates the isolation of AMF from the produced mixture allowing purities of up to 99.9% to be achieved.

Both HMF and AMF can be used to produce FDCA, with the advantage of using AMF being that it avoids the aforementioned issues associated with the HMF platform chemical while achieving an 82% yield. AMF also lacks the cytotoxicity and mutagenicity of HMF in humans [17]. This alternative addresses several of the obstacles to industrial HMF production. AMF shares the features of HMF that give it versatility as a platform chemical and can achieve the synthesis of 2,5-bis(hydroxymethyl)furan (BHMF), 5-hydroxymethylfuran-2-carboxylic acid (HFA), and FDCA (Figure 2).

### 2.2. FDCA from Hemicellulose

The other important platform chemical which can be obtained by acid-catalyzed hydrolysis of lignocellulosic biomass is furfural [13]. Produced from the xylan or hemicellulose contained in lignocellulose, furfural is a bulk chemical produced on a scale of 280,000 tons per year. Thus, the production of FDCA from furfural as a platform chemical may be more attractive than that from HMF, which has not yet been proven to be produced on a scale comparable to furfural [11]. This section presents three potentially valuable routes to FDCA from furfural.

Carbonylation is an attractive entry into FDCA production. It relies on 5-bromofuroic acid which is produced through the bromination of 2-furancarboxylic acid, a furfural oxidation derivative, which is currently used in the pharmaceutical industry as a process intermediate. It is a suitable platform for producing furfural-derived monomers as it is already well-established and hence more readily available than HMF [18]. Although the homogeneous carbonylation of 5-bromofuroic acid is possible, Shen et al. proposed a heterogeneous carbonylation method using a supported palladium catalyst to avoid the drawbacks of the homogeneous process and achieved a 97% isolated yield (Figure 3). The activated carbon-supported Pd(Xantphos)Cl_2_ catalyst that developed showed no significant decrease in performance when tested over 10 cycles [8].

With only NaBr as a by-product (which can be recycled after accumulation), this process demonstrates another successful alternative method of synthesizing bioderived FDCA, without the drawbacks of conventional HMF oxidation [8].

Interestingly, 2-furancarboxylic acid can also be directly carboxylated to FDCA using CO_2_ or inorganic carbonates. Dick et al. developed a process of producing FDCA by this method with an isolated yield of 89%, using a molten salt of cesium carbonate (Cs_2_CO_3_) (Figure 4) [19]. Although this method has shown good results on a laboratory scale, the industrial scalability of this process would be economically limited by the relatively high cost of cesium carbonate.

Alternatively, Nocito et al. proposed the synthesis and characterization of an alternative intermediate complex, copper-difuroate [20]. This option is more economical, as copper is cheaper than cesium. This new complex also increases the yield of FDCA up to 99%, compared to 76% starting from 2-furanoic acid under the same conditions due to the increased acidity of the proton in the fifth position on the furoic moiety, which allows for greater reactivity with the carbonate anion [20]. This is another potential route for obtaining FDCA from 2-furancarboxylic acid; however, the added complexity of the required expensive catalyst or intermediate complex may outweigh the benefits of eliminating the issues of the much simpler conventional oxidation of HMF. Further work is needed by research groups to determine the industrial feasibility of this method. A further option is the disproportionation of 2-furancarboxylic acid to FDCA and polyester. Polyesters are one of the main applications of FDCA. Pan et al. propose a method of producing FDCA directly from furfural along with 1,4-butanediol (1,4-BDO), which is polymerized with FDCA to produce poly(butylene 2,5-furandicarboxylate) (PBF).

As shown in Figure 5, this method first involves the oxidation of the furfural feedstock into 2-furancarboxylic acid. Catalytic aerobic oxidation is suggested for this [21]. The monoacid is then catalytically disproportionated into furan and FDCA by a variation of the Henkel reaction. Furan is converted to 1,4-BDO preferably through direct hydrogenation, but also possibly by catalytic oxidation to maleic anhydride and subsequent hydrogenation to 1,4-BDO through butyrolactone. Polycondensation can then be used to produce PBF. This method not only successfully produces FDCA from furfural but also produces an aliphatic diol that can be used to obtain a bio-based furanic polymer. As will be discussed in the next section of this review, polymerization with diols is FDCA’s main application. Most polymer synthesis processes utilize an aliphatic diol (such as 1,4-BDO) from one source and FDCA from another. If both components of polymerization can be synthesized from the same source of the platform chemical, it significantly simplifies process feedstock economics and increases the carbon utilization of the lignocellulosic biomass. Although only proven at the lab scale to date, this route has significant potential if it can be scaled up to commercial volumes.

## 3. FDCA as a Monomer

### 3.1. PET vs. PEF

PET is a fossil-based polymer widely used in food and beverage packaging, with over 50 megatons produced annually [22]. A bio-based furanic substitute for PET is PEF, most commonly produced via polycondensation of FDCA with ethylene glycol (Figure 3). PEF and PET exhibit similar thermal and mechanical properties. In terms of gas barrier properties, PEF outperforms PET, being 31 times less permeable to carbon dioxide. This makes PEF an excellent candidate for soft drink packaging applications [23].

Table 1 compares some of the thermal and mechanical properties of PET and PEF. The higher glass transition temperature of PEF results in a slightly more stable polymer under ambient conditions. A lower melting point reduces the cost of heating in downstream melt processing, improving the economics of industrial-scale production. The higher Young’s modulus and tensile strength indicate that mechanically superior plastic can be produced from PEF.

With similar behavior at high temperatures, both PEF and PET are reported to have high thermal stability up to 350 °C. The superior mechanical properties (including yield stress) of PEF are attributed to the additional motional constraints of the polymer. At a sufficiently high molecular weight, PEF is ductile with elongation at break values of 450% [23].

### 3.2. Polymer Production Challenges: Thermal Degradation and Stability

A common challenge encountered in PEF synthesis is the thermo-oxidative degradation and discoloration of the polymer during processing [24]. PEF is most usually synthesized via polycondensation, in which the reaction is limited by diffusion due to the high viscosity of the mixture. This results in long (i.e., up to 8–10 h) residence times in the reactor. Maintaining the mixture at reaction temperatures of around 200 °C during this time requires a significant amount of energy and increases the thermal degradation and coloration [25]. The result is an expensive industrial process that yields a product unsuitable for bottle packaging. In order to understand which methods will be most suitable for avoiding this issue, it is first necessary to understand the mechanisms of polymer degradation and the dependence of polymer choice on thermal stability.

Pyrolysis gas chromatography and mass spectrometry (Py-GC/MS) can be used to detect and identify the products of polymer degradation. Studies by Terzopoulou et al. used this method to characterize these products by degrading FDCA-based polymers made with aliphatic diols with varying numbers of methylene units (2, 3, 4, 5, 6, 8, 9, 10, and 12) [26,27,28]. At low pyrolysis retention times, volatile products such as CO, CO_2_, and H_2_O are vaporized first. The products formed at higher retention times are used to provide insight into the mechanisms of polyester degradation, which is important if stabilization methods to inhibit degradation are to be implemented. The degradation mechanisms of the FDCA-based polyesters are found to be identical and independent of the aliphatic diol used. The only effect of the number of methylene units on the degradation mechanisms is the specificity of the products formed (identical functionalities and mechanisms, different carbon chain lengths). Several polymer degradation mechanisms can be involved.

The primary decomposition mechanism for all FDCA-based polyesters is β-hydrogen scission on the ester bond. This produces vinyl- and carboxyl-terminated molecules, which were observed in all the polymers. The exact compounds differed depending on the number of methylene groups in the diol used for polymer synthesis. The β-hydrogen scission mechanism is common in polyesters. Double β-scission is also suggested to be the cause of the dienes found in pyrolysis gases [28].

Aldehyde-terminated molecules were also detected in the pyrolysis gas in smaller amounts. These are produced from polyesters by α-hydrogen scission or homolytic scission. These products were consistently found in considerably lower quantities than vinyl- and carboxyl-terminated molecules, which suggests that α-hydrogen is a less favorable mechanism than β-scission. In addition to these two pathways, other mechanisms can be operative. Small quantities of hydroxyl-terminated molecules were observed, which can be produced via the hydrolysis of the products of β-hydrogen scission [27]. Carbonyl derivatives of furan were also recorded, which can be formed via radical scission of furanoate or furfural, forming OH, H, or 2-furancarbonyl radicals.

Although the degradation mechanisms of FDCA-based polyesters remain the same with varying aliphatic chain lengths, there are slight differences in thermal stability. Although degradation occurs over a temperature range, Figure 4 shows the change in temperature at which the maximum degradation rate takes place for the polymers. The first data point in Figure 4 is PET, which degrades at slightly higher temperatures than FDCA-based polymers, indicating that it is slightly more thermally stable at high temperatures. It is important that any polymer intended to replace PET be able to achieve thermal stability as close as possible to its petroleum analog. It can be seen here that for short-chain diols (C2 to C4), increasing the chain length decreases thermal stability. This is because as the chain gets longer, the polymer becomes more flexible and less crystalline. A minimum is reached in the PBF. Then, the stability starts to slightly increase with increasing chain length. This is most likely due to the ability of longer polymer chains to fold and become more crystalline.

### 3.3. FDCA-Based Polymer Production

Beyond the stability issues of FDCA-based polymers, another challenge facing polymer production is the instability of the FDCA itself, which is expensive and discolors over time if the quality is poor. Many polymer manufacturers include stabilizers such as UV or antioxidants as additives to prevent coloration and polymer aging [24]. Furthermore, FDCA is poorly soluble in organic solvents and has a high melting point (over 300 °C), making melt polymerization processes difficult [29].

To address the instability of FDCA and prevent the thermo-oxidative degradation of FDCA polymers at high temperatures, the manufacturers DuPont and ADM esterify the FDCA they produce to its dimethyl ester, Me_2_-FDCA (Figure 6). This ester has a lower boiling point (112 °C) and is more readily soluble in organic solvents [29].

DuPont claims that synthesizing the dimethyl ester is more efficient and economical than simply producing and selling FDCA, as Me_2_-FDCA allows easier purification in order to obtain a polymer-grade product, is comparatively more stable during storage, and transport, and is advantageous for polymer manufacturing [30]. Me_2_-FDCA is reacted with bio-based 1,3-propanediol via polycondensation to yield a product that DuPont calls polytrimethylene furandicarboxylate, a polymer we will refer to as poly(propylene 2,5-furandicarboxylate) (PPF) in this review for the sake of naming consistency. DuPont claims that, like PEF, PPF has superior gas barrier properties to petroleum-derived PET [31]. The less harsh processing conditions of transesterification via Me_2_-FDCA compared to the direct polycondensation of FDCA to polymer, prevent thermo-oxidative degradation reactions and coloration of product.

Rosenboom et al. have developed a different process as a solution to the degradation problem. Their process involves prepolymerizing Me_2_-FDCA and ethylene glycol into linear PEF oligomers (Figure 7). These then form cyclic oligomers of varying sizes, which are polymerized by ring-opening polymerization to PEF, which meets bottle-grade specifications (i.e., colorless, high molecular weight of >30 kg mol^−1^, and >95% conversion).

Process conditions were varied to identify optimal conditions at 260 °C to synthesize bottle-grade PEF in less than 30 min. The researchers who developed this process reported the results of a preliminary economic evaluation indicating that an integrated ROP process could have a similar cost to the conventional polycondensation process. Thus, industrial ROP of PEF could be a more economically viable route than conventional polycondensation due to the higher quality of the final product and the reduced downstream processing required.

### 3.4. FDCA Copolymers beyond PEF and PPF

Beyond tuning the process and monomers to make FDCA polymers, copolymerization is another widely used method for imparting new and improved properties to polymers and tuning the parameters to create the most ideal material for a specific application. Therefore, it is interesting to examine the potential application of this method to FDCA-based polymers [32].

As with the other FDCA polymerization methods reviewed, melt polycondensation is the most commonly used method for copolymer synthesis, with the highly stable dimethyl ester of FDCA, Me2-FDCA, as the most popular starting monomer. The drawback of this method is the high reaction temperatures and residence times, which are highly energy-intensive and lead to the degradation of the polymer product. This issue prevents polycondensation from being used successfully on an industrial scale [32]. One solution to this issue is again through ring-opening polymerization, which offers shorter residence times and overall milder reaction conditions that avoid polymer degradation. Numerous studies report that copolymers synthesized by ROP yield higher molecular weights than when synthesized by melt polycondensation [32]. Another alternative method to melt polycondensation with milder reaction conditions is enzymatic polymerization. In addition to these milder conditions, enzymes also exhibit high selectivity for substrates and routes, generating fewer side reactions and by-products (economically advantageous). However, this high selectivity can also be a drawback as low yields will be obtained if a given substrate is unsuitable for the chosen enzyme [32]. For example, the common polymerization enzyme *Candida Antarctica* Lipase B (CALB) was shown to prefer longer chain diols, achieving the highest degree of polymerization using 1,8-octanediol [33]. Finally, another proposed method is reactive blending. By heating a polymer blend above the melting temperatures of all components, transesterification reactions and chain scissions in the reactive mixture then form the required copolymers [32]. Thus, we have here several potential routes to produce FDCA copolymers that can provide better quality products than conventional FDCA-based polymers, avoiding the thermo-oxidative degradation reactions that ruin polymer production. The next question that needs to be answered is the extent of the value that can be achieved by FDCA copolymerization.

Glass transition temperature (*T*_g_) is an important property to adjust for polymers. The *T*_g_ of a copolymer often lies between the glass transition temperatures of the two homopolymers used. This is more the case for non-crystalline random copolymers, where the *T*_g_ of the amorphous material depends directly on the monomer feed ratio [32]. FDCA as a monomer imparts high *T*_g_ to its polymers. In the synthesis of furanic copolymers, the incorporation of FDCA into the macromolecular structure increases the *T*_g_ in almost all cases studied [32,34]. The only exception to this phenomenon occurs when a bifuran diester is used as a comonomer, as bifuran rings have much lower mobility than furan rings [35]. *T*_g_ can also be tuned by other methods. The use of cyclic diols to prepare FDCA copolymers increases the *T*_g_ by impeding chain mobility [34]. Conversely, long-chain aliphatic diols increase chain mobility and lower the *T*_g_ of the copolymer [32]. Another important property to address is the crystallinity of copolymers. Many studies have been conducted on the block and random FDCA copolymers and how adjusting the homopolymer feed ratio changes the crystalline properties [32]. In the case of block polymers, the rigid furanic polymers provide the hard segment and are combined with various polymers for the soft segment. The most commonly used “soft” polymer is poly(ethylene glycol) (PEG). Increasing the PEG content in copolymers has been shown to increase the enthalpy of fusion (and thus the degree of crystallization) and crystallization rates [32]. A specific example is the introduction of increasing ratios of PEG to PBF to produce copolymers with increasing crystallinity due to improved chain mobility facilitating chain packing, as previously observed when increasing other semicrystalline polymer aliphatic chain lengths [36].

The final thermal property of interest here is thermal stability, which determines how a copolymer can be handled during processing and the applications for which the final product can be used. Dependent on many factors such as molecular weight and crystallinity, thermal degradation can also be significantly affected by the tuning of the chemical composition, which in turn is a function of homopolymer selection and feed ratio [32]. The reported thermal degradation temperature range for PEF-based polymers has been explored throughout various studies [32]. PEF has a considerable range from around 339 °C to 376 °C. The numerous copolymers of PEF cover an even wider range. Therefore, adjusting the chemical composition of copolymers is an effective and promising way to tune the thermal stability of polymers [32]. Furthermore, copolymerization at varying furanic polymer aliphatic diol methylene units (i.e., C2 to C6) provides a wide range of thermal stability. Considerable tunability can be observed in a range of FDCA polyesters using cyclic diols, carboxylic acids, and acyclic and a-hydroxy acids [32]. Thus, copolymerization is an attractive alternative method of producing furanic bioplastics. They can be synthesized using processes that are less harmful to the product than conventional polymerization, and their thermal properties can be customized to produce an optimized polymer.

## 4. BHMF as a Monomer

Another interesting route to FDCA valorization to be considered is the use of the diol of FDCA: 2,5-bis(hydroxymethyl)furan (BHMF). BHMF is an interesting bioderived compound that allows to obtain new polymers in areas such as self-healing polymers, resins, and crown ethers. There are several possible routes for the synthesis of BHMF from lignocellulosic biomass: (1) BHMF can be synthesized by the conversion of 5-hydroxymethyl furfural (HMF) by the Cannizzaro reaction. This occurs first by the formation of tetrahedral intermediates from the nucleophilic addition on HMF between the carbonyl group and the hydroxyl group. The hydride is then transferred to a carbonyl carbon to create BHMF and HMFA [37,38]. Since two products (BHMF and HMFA) are generated in relatively high yields, the commercial application of this route requires sufficient separation technology. (2) BHMF can also be synthesized by the catalytic hydrogenation of HMF. A few reports also discuss the potential of catalytic transfer hydrogenation reduction using formic acid, methanol, ethanol, and isopropanol as hydrogen donors [37].

### 4.1. BHMF-Based Polymer Production

This section will review the many polymers available from BHMF, focusing on polyesters, polycarbonates, and polyurethanes. In addition to these three types of polymers, it is important to note that BHMF can also be polymerized with succinic acid to produce self-healing materials that are capable of self-repair due to the reversibility of the relatively weak dynamic bonds on their surface. These polymers are produced by esterification between BHMF and succinic acid, and subsequent Diels–Alder reactions with 1,8-bis(maleimido)-triethylene glycol. Increasing the BHMF content was found to increase the healing ability of the resulting polymers [39]. The rigidity of the furan ring also makes BHMF an excellent candidate for the synthesis of epoxy resins. BHMF-based resins have superior thermodynamic properties and storage moduli as well as lower viscosity than the conventional phenyl-based alternatives [40].

The BHMF diol can be reacted with diacid ethyl esters to produce a range of polyesters with interesting properties. As with FDCA-based polymers, the variation in the number of methylene units on the non-furanic monomer can affect the properties of the BHMF-based polymer. To this end, Loos produced BHMF-based polymers using diacid ethyl esters with a varying number of methylene units (Figure 8) [41]. In this scheme, the notation ‘x’ indicates the number of methylene units of the diacids, which for this study included: 2 (succinate), 3 (glutarate), 4 (adipate), 6 (suberate), 8 (sebacate), and 10 (dodecanedioate). To achieve this polycondensation, enzyme catalysis with CALB was implemented. This enzyme has been widely studied in polymerization biocatalysis due to its ability to produce precise polymeric structures without the toxic residues generated by conventional catalysts [42,43]. This “green” enzymatic polycondensation pathway involved the synthesis of the oligomers of BHMF and diacid ethyl esters by CALB in two steps, followed by a final step of condensation of the oligomers into polyesters [41]. However, the molecular weights of the polymers synthesized in this study remained below 2 kg/mol. This is proposed to be due to the formation of ether linkages. The highly reactive hydroxyl groups of the BHMF diol can indeed react with the ethanol by-product or even dehydrate together to form BHMF ethers. These side reactions that decrease the molecular weight could be avoided by reducing the reaction temperature or decreasing the residence time. The valorization of these BHMF ethers into high-value products is considered as a method to make industrial BHMF-polymer production more economically viable (*vide infra*).

Wide-angle X-ray diffraction analysis was used to determine the degree of crystallinity of the BHMF polymers (Figure 5). An increase in the degree of crystallinity from 34–65% is observed as the methylene content of the diacids increases from 2 to 10. In the “conventional” FDCA polymers studied, beyond PBF, the increase in the methylene content in the aliphatic diol provided more chain flexibility, allowing further chain packing, which overall increased the crystallinity. The same phenomenon is observed here, as longer diacid chains generate an overall trend of increased crystallinity [41]. Thermogravimetric analysis (TGA) of the polymers provided insight into how the number of methylene units affected the stability and degradation kinetics of the polymers. Two distinct decomposition steps were observed. Around 75–80% of mass loss occurred at 276–332 °C, followed by 11–16% loss at 436–453 °C [41]. This is slightly lower than the thermal stability of FDCA-based polymers but still relatively high. The temperature at which degradation began and reached its maximum rate consistently steadily increased with the increasing number of methylene units, similar to FDCA polymers beyond PBF, further validating that furanic polymers with longer methylene chains are more thermally stable than their shorter chain counterparts [41]. The observed glass transition temperatures with different diacids were more interesting. As shown in Figure 6, the *T*_g_ starts relatively high, indicating high chain rigidity. However, as the number of methylene units increases from 2 to 6, the *T*_g_ drops from 4 to −38 °C. Beyond that, the *T*_g_ then increases to −8 °C as the number of methylene units increases to 10 [41].

This behavior is similar to that observed for the thermal stability of FDCA polymers. As the polymer chain length increases from the smallest polymer, the flexibility and mobility of the chain increases. As the polymers become even longer, the chain can then pack and fold and thus becomes more crystalline. This behavior is observed in most semicrystalline polymers, where crystallites reduce the ease of molecular motion to increase *T*_g_ [44]. Thus, thermally speaking, BHMF polyesters exhibit similar behavior to the FDCA polyesters. The advantage of these polymers is the environmentally friendly nature of the enzymatic production method compared to conventional FDCA polymerization.

An alternative attractive pathway to BHMF-based polyesters from FDCA was recently introduced by Thomas and Gauvin relying on one-pot catalysis [45,46]. Thus, an esterification-hydrogenation-copolymerization sequence was developed using molecular catalysis, enabling the one-pot transformation of FDCA into the desired polyesters. These were synthesized with *M*_w_ values up to 16,000 g/mol for the (BHMF-adipic acid) copolymer (Figure 9). Compared to a similar strategy starting from HMF, this path offers the benefit of being free of humin traces, a key point in the view of their potential application.

Polycarbonates (PCs) are an important group of polymers known for their strength, stability, and transparency. Traditionally, PCs are synthesized by coupling a diol with the highly toxic phosgene. Potential alternatives to phosgene include dimethyl carbonate and diphenyl carbonate. In addition to lower reactivity than phosgene, these materials require the separation of by-products at high temperatures to maintain reaction kinetics. This presents the same issue as FDCA-based polymer production: if BHMF is to be used as a diol, these high temperatures will result in thermo-oxidative degradation due to the low thermal stability of BHMF [47]. Some research groups have proposed fluoride-triggered carbonylation to produce PCs via carbonyldiimidazole (CDI). This eliminates the need for high temperatures or the removal of by-products. CDI can also be converted from diphenyl carbonate. The aliphatic PCs produced are biodegradable but suffer from low glass transition temperatures. The application of BHMF, in this case, presents an opportunity to improve these polymers, as the furanic moiety will increase chain rigidity and hence *T*_g_, which is highly dependent on polymer chain torsional freedom. Choi et al. have developed a method using the Diels–Alder reaction between BHMF and a dienophile to produce a rigid bicycle that can decrease the torsional freedom of PC polymer chains and increase *T*_g_ [47]. Their two-step CDI-mediated method was accomplished under mild conditions (required for BHMF thermal instability) due to the high polymerization reactivity of CDI. Excess CDI is reacted with BHMF to convert the BHMF hydroxyl groups into imidazole carboxylate groups for subsequent polymerization, generating a furanic polycarbonate. Cesium fluoride was used as a catalyst. The reaction temperature was varied to obtain a clear polymer with *M*_w_ = 59 kg/mol at 40 °C.

TGA indicated that the resulting PC was thermally stable up to 150 °C. The Diels–Alder reaction with furan cycloaddition of the conjugated diene and dienophile was used to tune the glass transition temperature. Maleimide was used as the dienophile and with tuning, achieved *T*g = 80 °C [47]. This presents a more specialized application for BHMF. The rigid furanic diol requires mild processing conditions. A mild process exists for producing biodegradable PCs but it suffers from low *T*_g_. This method combines both issues to create a niche route for valorizing the low-thermal-stability BHMF to produce biodegradable polycarbonates with a high glass transition temperature.

Polyurethanes (PUs) are an important and versatile family of plastics that can be used as thermoplastics and thermosets in a range of applications, including adhesives, packaging, coatings, construction, and foams [48,49]. The general method of synthesizing polyurethanes is a polyaddition reaction of polyols and diisocyanates by catalysis or ultraviolet activation. This method comes with two problems: the toxicity of the phosgene used to produce diisocyanates [50] and the high reaction temperatures that are unsuitable for the thermally unstable BHMF [49]. To combat the issue of unsuitable conventional reaction conditions, Oh used high-speed vibrational ball milling to develop a solvent-free, mechanochemical method for the synthesis of BHMF PUs. This process is advantageous as the facile reaction mechanism avoids the use of biphasic solvents, which also reduces the production of process waste. This solid-state process was found to have a significantly shorter residence time than the conventional method. The diisocyanate and catalyst used, the vibrational frequency, and the ball mill residence time were modified to optimize the molecular weight obtained for the polymer. The authors concluded that the optimal method tested used methylene diphenyl diisocyanate with dibutyltin dilaurate as a catalyst, which yielded a PU of *M*_W_ = 163 kg/mol. This method resulted in a glass transition temperature of 96 °C and promising thermal stability (*T*_d,5%_ = 201 °C). The molecular weight and physical properties of the produced polymers were further modified by the addition of diols/diamines in the process to successfully produce furanic-*co*-polyurethane (FR-*co*-PU) copolymers in one pot.

To address the toxicity of the phosgene used to produce the diisocyanate for PU synthesis, Zhang developed a method to synthesize nonisocyanate polyurethanes (NIPUs). This method involves the transurethanisation of BHMF and dicarbamates. 1,4-butanediol was included to add hardness and flexibility to the NIPUs. The furan moieties were reacted with bismaleimide to crosslink the NIPUs via a reversible Diels–Alder reaction, providing a potential application in easily recyclable and mendable/self-healing polymers [50].

Although PUs are useful for a wide range of applications, there are many methods of surface modifications to impart specific additional features to the polymers, such as adhesion, conductivity, wettability, and catalysis. “Click” chemistry is often used for this by incorporating a diol with a “clickable” moiety as a building block of the PU, which allows various additional functional groups to be “clicked” onto the polymer, thus expanding the choice of PU properties and modification options from the same feedstock components. For instance, Nguyen proved the applicability of BHMF as a “clickable” diol component to allow the addition of methylmaleimide through a reversible Diels–Alder “click” mechanism [48]. This paves the way for future studies to determine the range and extent of additional PU properties available using BHMF for “click” chemistry. Polyurethanes then present a whole new range of suitable applications for BHMF. Together with polyesters and polycarbonates, BHMF appears to have a wider range of applications than the diacid FDCA.

### 4.2. BHMF Ethers

As discussed previously, the reactivity of the OH groups on BHMF can lead to dehydration side reactions during processing. The OH groups can react with the hydroxy groups of other BHMF molecules or with polymerization by-products, producing BHMF ethers and ether end groups (Figure 10) [41].

For this reason, variations in the chain length of the diol used in polymerization often have little to no effect on enzymatic polymerization. This side reaction affects the stoichiometric ratio of the OH groups and ethyl ester groups and hinders the growth of the polymeric chain where the ether end groups are formed. This results in low-molecular-weight polyesters unless BHMF ether formation is inhibited [41]. The specific ethers formed depend on the alcohol with which the OH group of BHMF reacts, but this general group of ethers can be referred to as 2,5-bis(alkoxymethyl)furan (BAMF). Although the inhibition of ether formation is a potential solution for increasing the molecular weight of BHMF polyesters, before any method recommendation it is important that the alternative has been considered. In this case, the alternative method would be to separate and valorize these BAMF ethers.

Similar to the replacement of PET with PEF, biomass-derived furanics can also be used to reduce the dependence of fuels on fossil sources. Current attempts to develop renewable biodiesel have mostly focused on the production of fatty acid methyl esters by the transesterification of triglycerides and alcohols. Although these biofuels are efficient and can easily be used in modern engines, they also pose several problems. These include high feedstock costs, low stability, poor flow properties, and the negative implications of using human food sources as fuel [51]. Again, lignocellulosic biomass presents an interesting alternative route to replace a field dominated by fossil fuels. BAMF compounds are relatively stable materials with low freezing points and high cetane numbers, making them well suited for use in diesel additives. Several research groups have focused on synthesizing BAMF from HMF using an amberlyst-15 catalyst for this purpose [52]. Cao et al. identified suitable BAMF compounds, with 2,5-bis(methoxymethyl)furan (BMMF) and 2,5-bis(ethoxymethyl)furan (BEMF) presenting attractive options due to the simplicity of acquiring the necessary alcohols required for etherification [53]; methanol being the cheapest alcohol and ethanol presenting further opportunities to achieve a circular and sustainable process due to the ability to source the alcohol from renewable cellulose or starch. The authors reported relevant properties of BMMF regarding diesel improvement, including solubility in diesel, a boiling point of 190 °C, a high flash point of 90 °C, and a low cold filter plugging point of less than −37 °C. Most important was the cetane number, which is a measure of how readily combustible a fuel is. Commercial diesel often has a cetane number between 45 and 50. BMMF was found to have a cetane number of 80. Therefore, the addition of BMMF to diesel would certainly improve the quality and combustibility of the fuel [53]. In 2011, Avantium had planned to apply the products of their YXY process as biofuels, before later shifting the business model to bioplastics. At that time, they conducted European Standard Cycling (ESC) testing of their “YXY fuel components” on diesel engines to determine how varying mixtures of different furanic compounds would affect engine performance. This work tested BEMF and found no significant difference in engine operation, implying that little or no modification is required for modern diesel engines to accept the addition of BAMF additives into commercial diesel streams [54]. Although studies in the literature provide a potential future route for valorizing the BHMF ether by-products formed during BHMF polymerization, most research currently remains focused on perfecting the catalytic synthesis of ethers, with little work on diesel streams. Furthermore, the literature does not provide insight into why Avantium shifted its business model to bioplastics over the past decade—specifically, whether bioplastics were chosen as they are more economically attractive, or whether they determined that the biofuel route was currently unfeasible. Either way, BHMF ether valorization is an intriguing route if ether formation cannot be economically inhibited during BHMF polyester production.

## 5. Industrial Production and Economics

The industrial production of FDCA is still very new, with a few promising manufacturers and processes moving to pilot scale in the coming years. This section will review some of the relevant advances in industrial technology, consider the methods for improving the economics of the industrial process, and review economic and business considerations and the competition for FDCA production.

### 5.1. Advances in the Industrial Production of FDCA and Esters

Originally a “spin-off” project of Shell in 2000, Avantium is on track to become the first company to successfully produce FDCA on a commercial scale [55,56]. Avantium has been operating a pilot plant in Geleen, the Netherlands, since 2011. This plant successfully uses Avantium’s YXY^®^ Technology, which the company claims to be “the most advanced production technology for PEF across the sector”. The technology involves the catalytic dehydration of fructose to alkoxymethylfurfurals, followed by its oxidation to FDCA [57]. This process is similar to the conventional HMF oxidation process discussed earlier. Avantium’s RAY Technology^TM^ is currently being successfully operated on a pilot scale to catalytically convert sugars into ethylene glycol. This technology is being scaled up in a demonstration plant in Delfzijl, the Netherlands [55]. The produced FDCA and ethylene glycol are catalytically polymerized to produce poly(ethylene furanoate) (PEF). The current capacity of Avantium’s pilot FDCA plant is 40 tpa (tons per annum) [58]. In 2019, the company acquired Synvina from a joint venture with BASF and renamed the business Avantium Renewable Polymers, which has been granted EUR 25 million to build a new “flagship” plant [55]. Avantium expects this plant to have a capacity of 50,000 tpa of FDCA and to begin commercial production in 2023 [57].

AVA Biochem has successfully piloted a continuous water-based process for producing HMF and has been commercially producing high-purity HMF since 2014 [58]. Their patented AVA hydrothermal carbonization process involves the hydrolysis and dehydration of biomass before subsequently polymerizing intermediate products into large biocoal molecules. One of the intermediates is HMF, which can be extracted prior to polymerization during continuous production. The extraction product is crystallized HMF with a purity of up to 99.9% or HMF in an aqueous solution [59]. AVA Biochem also claims to have started production of FDCA with a capacity of 30,000 tpa and plans to increase it to 120,000 tpa [56]. AVA sells some of the produced HMF and also uses it to produce and sell highly pure DFF. The FDCA produced is used to synthesize PEF and the dimethyl ester of FDCA for sale [60].

The process used by DuPont and ADM to produce Me_2_-FDCA and PPF has been discussed in more detail in the previous section. PPF has potential applications in food and beverage packaging, which requires polymers to be of high quality and colorless. The starting platform chemical, bio-based HMF, is often synthesized with polymeric impurities such as humins, which add to the dark yellow coloration of feed and other derived products [31]. It is, therefore, necessary to remove humins from these Me_2_-FDCA-producing processes if the resulting polymers are to be used in the packaging industry. DuPont and ADM have co-developed a process for producing Me_2_-FDCA from HMF to a “substantially free of humins” end product. This patented process involves the oxidation of bio-based HMF to produce a mixture of humins and FDCA. This mixture is then esterified with methanol to obtain Me_2_-FDCA. The product is then purified (i.e., separated from humins) via distillation or sublimation [31]. The DuPont ADM Me_2_-FDCA program has been proven on a pilot scale, and a 60 tpa commercial development plant is currently in operation in Decatur, Illinois. DuPont claims to be on the path to the commercialization of this process [61].

Origin Materials, which is a new entrant in the FDCA field, catalytically converts lignocellulosic biomass into four platform chemicals, namely levulinic acid, hydrothermal carbon, furan, and CMF [62]. As previously discussed, CMF and furfural are key platform chemicals in the production of FDCA. Origin recently acquired a process from Eastman Chemicals to commercially produce high-purity FDCA [63]. This technology is being applied to build a pilot FDCA plant in Sarnia, Canada [64].

### 5.2. Improving FDCA Process Economics: Humin Valorization

Commercial HMF and FDCA production is largely hampered by the synthesis of humins. These by-products often appear as an insoluble, black, tar-like material during bioprocessing. Complex and poorly understood, the mechanism of humin production is thought to arise from condensation reactions between carbohydrates (sugars) and intermediates along the pathway to platform chemicals such as HMF (Figure 11) [65,66,67]. The structure of humins is also poorly understood. They are considered to be highly functional oligomeric bio-macromolecules. The polydisperse furanic molecules contain aldehyde, ketone, hydroxyl, and carboxylic functionalities [66]. Currently, humins are most often discarded as waste products. Since humin by-products are produced with considerable selectivity, this generates large expenses on an industrial scale. A new valorization of humins into value-added products is, therefore, an important potential route to make large-scale biorefineries more economical.

Humin-based thermosets and composites for building materials are one of the potential application fields for these species. Impregnation of cellulosic matrices with furanic resins can create bio-based composites for construction such as decking, flooring, boards, or plywood. Humins can be incorporated into polyfurfuryl acid (PFA) and impregnated into cellulosic matrices to produce biocomposites with excellent mechanical properties [68,69]. This increases the renewable carbon content of building materials and reduces costs, as humins are a cheap by-product of biorefinery [65].

Wood products for use in construction currently require impregnation with biocides, or modification to ensure dimensional stability and biodegradation resistance. Modification is also required to reduce the combustibility of the wood products [66]. Similar to the previously mentioned resins, furfuryl alcohol (FA) is currently used to modify wood. FA is injected into the wood and polymerized within the cell walls into PFA. The furfurylated wood product is thermochemically superior to the untreated wood, making it more suitable for construction [66]. Humins are very similar to PFA, and hence present a suitable alternative for wood modification. Sangregorio et al. have shown that humin-modified wood exhibited similar thermochemical and mechanical properties to PFA, which are necessary to meet the criteria for this application [66]. This again increases the value of humins by valorization and reduces the content of the PFA used (and hence the cost of the modification process).

Catalysis is a further domain in which the implementation of humins has been probed. Indeed, catalysts used for selective oxidative hydrocarbon cleavage are often toxic or environmentally harmful. As this is such an industrially relevant process, there is great interest in finding suitable “green” catalysts [70,71]. Abundant, inexpensive, and environmentally benign, iron-based catalysts are promising green catalysts for these processes and show good results. Filiciotto et al. demonstrated that humins can be used to produce iron oxide catalytic nanocomposites using a safe and environmentally friendly process. The composites were applied to the synthesis of vanillin by the microwave-assisted selective oxidation of isoeugenol and achieved conversions >87%. This catalyst was found to be highly efficient and reusable, providing a promising valorization route for humin by-products.

As a final example, the high carbon content of humin by-products (60 wt% C) offers good potential for gasification to synthesize gas (syngas) or hydrogen [67]. Hoang et al. proved that gasification of humins produced a graphitic carbon by-product and a volatile organic gas mixture of phenols, aromatics, and furans, which can be used as feedstock for syngas or H2 production. The process has been improved using a sodium carbonate catalyst [67].

### 5.3. FDCA Market Economics

The development of biorefineries based on new technologies and products faces significant economic challenges. Arguably the greatest challenge for companies, which are aiming to produce poly(ethylene) furanoate industrially and commercially, is market adoption. The price of their product must cover the cost of the production (with new methods) and the investment in production plants and infrastructure. These prices have to remain similar to (or ideally lower than) the price of the product that they are trying to replace. This is difficult when the product to be replaced is a well-established large-scale fossil plastic, whose production process has been optimized over decades. This section will cover an analysis of the target production costs for an FDCA/PEF production plant to be profitable based on slightly different production methods, considering economies of scale and a sensitivity analysis. It will also examine Avantium, the current leader in FDCA commercialization, its current plans to address the market risks associated with large-scale FDCA/PEF production and compare this to DuPont and Corbion, other competitors.

Researchers at the Copernicus Institute for Sustainable Development, in partnership with Avantium, have conducted technical and economic studies of three cases of FDCA production processes based on a new technology [72]. As previously discussed, Avantium’s Dawn technology converts lignocellulosic biomass into industrial sugars such as alkoxymethylfurfurals [73]. Avantium’s Ray technology converts similar plant-based materials into ethylene glycol [73]. The alkoxymethylfurfurals are converted to FDCA and combined with ethylene glycol to produce PEF using Avantium’s YXY technology [73]. This is the basis of all three cases, with slight process variations. The main difference between cases lies in the technology applied to the pre-treatment of the lignocellulosic feedstock: Case I: Organosolv fractionation with ethanol; Case II: Organosolv fractionation with methanol; Case III: Furasolv fractionation with methanol.

Based on the underlying experimental data, the different pre-treatment methods account for the varying yields of the products used in each case. The products obtained include PEF, furfuryl ethyl ether (FEE), dimethyl ether (DME), and methyl levulinate [72]. The researchers determined the capital investment costs using ASPEN Plus Economic Analyser, and the PEF production costs using net present value (NPV). The costs of the raw materials and the selling prices for the product lines were based on the market averages for their applications. FEE can be used as an industrial solvent or as a biofuel, so prices were obtained for both applications and used based on a production scale. DME was also priced based on its use as a biofuel. ML was priced based on its current small-scale application as a pharmaceutical intermediate, as well as a potential biofuel and bisphenol-A replacement [72]. The researchers found that the price of the petroleum analog of PEF, PET, ranged from 1500 USD/ton to 2300 USD/ton, with an average price of around 1800 USD/ton [72]. Therefore, the basis for a marketable production price for PEF is 1500 USD/ton. If a process is capable of synthesizing PEF at or below this cost per ton, then it can be profitably sold in today’s market as a PET replacement. The total value of the capital investment required does not vary significantly between cases, ranging from USD 346 million to 353 million. The only notable difference between cases is the distribution of the total direct costs, for which different areas of the process take a larger share depending on the different methods employed [72].

To understand the economies of scale of these processes, the annual feedstock throughput was varied to 100 ktpa, 625 ktpa, and 1 Mtpa. The PEF production cost as a function of biorefinery size was determined for each case. Remembering that the goal is to achieve a competitive market price of 1500 USD/ton or less, this allows for the evaluation of cost-effective scales. The production costs were determined by using an NPV equation and setting the value of NPV to zero, which indicates the PEF selling price that would achieve the break-even point and balance the production costs. Economies of scale are noticeable when going from 100 ktpa to 625 ktpa. However, from 625 ktpa to 1000 ktpa, the shift in the application of FEE from industrial solvent to bio-based fuel additive neutralizes the scaling effects [72]. The scaling effects are much more relevant in Case I than in the other cases. Indeed, the production of PEF decreases steadily throughout the scales, from 80 ktpa to 52 ktpa to 22 ktpa. In Cases II and III, ML, DME, and electricity revenues increase more rapidly than PEF revenues, resulting in PEF playing a less significant role in total revenues than in Case I [72]. Sensitivity analyses were also carried out to produce spider diagrams for all three cases. Overall, there was a strong linear effect on the PEF production costs from the capital costs, discount rate, and feedstock costs. This provides insight into the key economic factors to which PEF production is sensitive, and thus the most important areas to optimize for an overall cheaper and more profitable product.

Overall, this economic analysis indicated that for the primary market application (PET replacement), PEF production costs are too high to compete at a low production scale. The large-scale processes, however, showed close competition with PET. With the added value of superior thermal, mechanical, and gas barrier properties, as well as attractive renewability and “eco-friendliness”, PEF could compete with conventional PET if produced on a sufficient scale. Finally, although extensive, this study does not consider the additional factors associated with the development of new technology such as the learning effects, risk financing, or the need for demonstration plants (cost and timescale effects).

Avantium is currently designing and planning a 50 ktpa FDCA/PEF flagship plant to begin commercial production in 2023. As mentioned earlier, PEF production at this scale is not economically competitive with PET on a cost-effective price basis alone [72]. Avantium’s business plan then relies on other market factors and applications. In addition to the potential to replace packaging materials, PEF also has applications in the film and fiber markets. Avantium claims that the end markets for these materials represent over USD 200 billion annually [74]. The new flagship plant will begin to enter the market for these high-value applications by partnering with niche users who require PEF’s unique features [74]: films for electronic displays (LCD/OLED), PEF-reinforced bottles for premium product packaging, and plant-based packaging. This gives Avantium the opportunity to increase its experience and knowledge of PEF production at the 50 ktpa scale while making profits in a small, high-value market. This then leads to a future where PEF production can be developed on a large enough scale to be competitive in the high-volume market of bottle packaging. The market entry plan also involves pricing plans based on market estimates of what will be competitive for specific applications. Initially, PEF for high-value products can be sold at 8–10 EUR/kg, then at a higher scale for medium-value products at 4–5 EUR/kg. Finally, the long-term goal is large-scale global production with >100 ktpa per plant, producing PEF at 1.5–2.5 EUR/kg [74]. This suggests that Avantium has recognized the impossibility of replacing PET with PEF at the current scale of production. This development plan calls for more niche applications to generate revenue at the current scale, whereas the technology is developed through experience to a scale suitable for replacing PET. For the long term, Avantium also relies on other market factors. PET bottle consumers drive the transition to bioplastics through their purchasing behavior, encouraging retailers and beverage companies to adopt circular bioplastics. Governments can also price carbon, mandate the use of sustainable products, and provide market incentives for innovative companies to generate green solutions [74]. As an example, an EU agency awarded Avantium a EUR 25 million grant to support the development of PEF and FDCA [74].

Although Avantium plans to cover the production chain from raw material to final application, other companies are making progress by focusing on smaller sections of the supply chain (Figure 7). As discussed above, DuPont has developed a process with ADM to produce the dimethyl ester of FDCA, Me_2_-FDCA, from fructose via HMF. Although they are conducting studies on polymerizing the ester with propanediol to produce poly(trimethylene furanoate), their current focus is on producing Me_2_-FDCA as a monomer for sale and application [61]. This is in contrast to Avantium, who focuses on a specific monomer (PEF). As different furanic polymers are tested and come to market, DuPont has more application flexibility as their product is not polymer-specific.

Corbion is a market leader in another bioplastic monomer, lactic acid. This company has also recently begun work on using biocatalytic processes to convert HMF into FDCA. Corbion’s process can achieve >99% yield of FDCA, which provides a high PEF yield per mass of feedstock if applied to polymer production. A distinct advantage of this process is the ability of Corbion’s biocatalytic microorganisms to process raw HMF feedstocks with various impurities (e.g., humins) and produce FDCA with virtually no by-products, in contrast to more conventional routes that require HMF impurity removal to avoid the discoloration of the FDCA product [75]. The risk Corbion faces is due to its lack of coverage/influence on the furanic polymer supply chain, as shown in Figure 7. Although companies like Avantium and DuPont can cover raw feedstock to the final product, Corbion is highly dependent and sensitive to fluctuations in the raw HMF supply and price, as well as the demand for FDCA from polymer manufacturers. Despite the efficiency of the FDCA synthesis process, Corbion’s narrow business model relies on HMF suppliers not wanting to expand their own businesses into FDCA production—which is exactly what Ava Biochem is working on. Already a leader in HMF production and having proven its process of electrosynthesis to FDCA on a small scale, Ava Biochem is moving toward a 3–5 tpa FDCA plant. AVA has received a EUR 50,000 grant from the EU Horizon 2020 to help achieve this goal [76]. It is clear that several companies are making interesting and unique advances in commercial FDCA production, and the next few years will test not only the chemistry and engineering of these processes but also the business models designed for market entry.

## 6. Conclusions

The conventional method of FDCA synthesis via HMF oxidation has two distinct disadvantages: humin by-product generation and low FDCA selectivity. If the valorization of cellulose is to be adopted, HMF and AMF present two platform chemicals for FDCA production. If HMF is chosen, electrocatalytic oxidation of the product mixture will be required if high FDCA content is to be achieved and the humins removed. AMF presents a better alternative to HMF without these issues but requires a less facile route and additional side reactions. Alternatively, hemicellulose from lignocellulosic biomass can be valorized to furfural, a commodity chemical. It has been shown that furfural can be converted to FDCA through a variety of routes. Manufacturers facing the low profitability of commercial HMF oxidation should consider developing these new alternative methods.

Classical FDCA-based polymer production is hampered largely by low thermal stability resulting in thermo-oxidative degradation during polycondensation reactions. The degradation has been understood to occur via β-hydrogen, α-hydrogen, and homolytic scission. Ring-opening polymerization is a method with high potential to avoid the high temperatures of conventional polycondensation and thus prevent degradation. The dimethyl ester of FDCA, Me_2_-FDCA, has been shown to be a better monomer than FDCA due to its higher stability during transport and storage, as well as its easier polymerization. Copolymerization has proven to be an attractive method for producing furanic bioplastics, with environmentally friendly methods allowing a high degree of tuning of the thermal properties to produce various polymers.

As an alternative to FDCA diacid and dimethyl ester, the diol BHMF has shown excellent polymerization potential. BHMF-based polyesters exhibit similar properties to FDCA-based polyesters, with the advantage of having a “greener” enzymatic polymerization method. BHMF-based polycarbonates present an interesting and potentially high-value niche application. BHMF is also capable of producing high-quality polyurethanes. Overall, although FDCA applications may be slightly more well-established, the diversity and range of applications available through BHMF make it a potentially more attractive monomer than FDCA. Furthermore, the BHMF ether by-products from BHMF-based polyester production can be valorized to generate revenue for commercial processes if etherification cannot be inhibited.

Finally, industrial FDCA/PEF production has been evaluated. The general application of PEF to food and beverage packaging is only viable at a sufficiently high production scale; smaller-scale commercial production will rely on high-value niche applications of PEF. Process economics can be improved by the valorization of humins. Nevertheless, several manufacturers are entering this new market and the coming years will test their engineering and business skills.

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
