# Peer review of "2,5-Furandicarboxylic Acid: An Intriguing Precursor for Monomer and Polymer Synthesis"

_molecules, 2022, doi:10.3390/molecules27134071_

Round 1

Reviewer 1 Report

The authors have nicely reviewed in detail several aspects of 2,5-furandicarboxylic acid (FDCA). This molecule indeed will revolutionize biopolymers/biochemicals in the near future. The authors have covered chemical aspects of FDCA with additions of industrial production and economics at the end. I recommend publication with only minor corrections shown below.

-       - There have been several reviews relating to FDCA and polymerizations. Authors should add more citations in the introduction so that readers can do further reading on the subjects.

-          - In section 3.2 on page 7, authors described thermal degradation and stability of PEF. This is a nice section but difficult to follow. A scheme or figure showing thermal decomposition mechanism (such as alpha and beta-H scission) would be very nice.

Minor typos:

-          Page 3. Figure 2 somehow appears twice. Please check.

-          Page 8, line 254. Typo maybe, FDC -> FDCA?

-          Page 14, line 446. Typo maybe, BHFM -> BHMF?

-          Page 17, line 572. Starting from this point this section should be #5, not #4.

-          Page 22, line 779. Typo maybe, a specific monomer -> a specific polymer?

Reviewer 3 Report

Marshall et al. reported new alternative methods to prepare FDCA building block. The review submitted is a very good work reporting a detailed bibliographic study on an extremely hot research topic nowadays. It is also a well-written manuscript.

I highly recommend publishing this work after addressing the following points:

- Introduction (page 1, line 29) : the sentence ‘The production of FDCA has been studied as early as the 18th century;’. Authors should add a relevant reference for this.

-Introduction (page 2, line 48-53) the sentence ‘Although some companies are taking steps toward commercial FDCA production, conventional synthetic routes still present many drawbacks that hinder industrial applications. In addition, the main application of FDCA for the synthesis of poly(ethylene furanoate) (PEF) has not been directly compared to the alternative polyesters, polycarbonates, and polyurethanes that can be produced from the diol of FDCA. In this review article we will present recent developments in the production and use of FDCA.’.

Authors should add references that confirm the following statement: “...conventional synthetic routes still present many drawbacks that hinder industrial applications”. Authors can specify which conventional synthetic routes that present drawbacks they meant? Which industrial applications they meant? This sentence is not clear. Better to reformulate it and to mention in detail those drawbacks.

I also wonder what is the significance of comparing PEF with alternative polyesters, polycarbonates and polyurethanes that can be obtained from the DIOL of FDCA? What is the connection??

Authors should reformulate this sentence.

- Page 3: Figure 2 is duplicated. Please remove one. Lines 78-88: the font used in this paragraph is different from that used in the rest of the manuscript. Please make the font consistent throughout the manuscript.

- Authors mentioned in page 3, line 100 that the electrolytic oxidation of HMF directly to FDCA showed low yields. But they reported nothing on the yield of obtaining FDCA starting from fructose or glycose by air oxidation of AMF (depicted in Scheme 2). For comparison purpose, authors should mention the yield of FDCA in Scheme 2 and add a relevant paragraph to highlight the efficiency of this pathway compared to other processes.

-Page 5, line 148-157; authors can add the yield, reported in the literature, of obtaining FDCA from xylose (Scheme 3). The same thing should be reported in Scheme 4.

-Page 5, line 168: please add the obtained yield of FDCA.

- Page 7, line 219-220: the sentence ‘This results in long (i.e., several days) residence times in the reactor.’ is not correct, stating that polycondensation reaction of PEF takes several days???. Contrarily, it does not usually take more than 8-10 hours maximum. Please adjust this statement and add relevant references.

 -Page 11, paragraph in line 339-340: the sentence ‘The use of cyclic diols to prepare FDCA copolymers increases the Tg by impeding chain mobility.’ Please add more relevant recent references confirm this statement (effect of cyclic diols ‘e.g. isosorbide’ to develop furan-based copolyesters featuring high Tg values). Many excellent works dealing with furan-based copolyesters have been recently published that confirm authors’ statement.

-Page 11, paragraph in line 335-337: the sentence “In the synthesis of furanic copolymers, the incorporation of FDCA into the macromolecular structure increases the Tg in almost all cases studied.” Authors can add more new references confirming that because they just mentioned one or maximum two references (mainly reference 29).

-Page 15, line 464-467: please add relevant references for the following sentence: ‘Choi et al. have developed a method using the Diels-Alder reaction between BHMF and a dienophile to produce a rigid bicycle that can decrease the torsional freedom of PC polymer chains and increase Tg.

I can recommend publishing this excellent work only after addressing all points I mentioned above.
